# A Learning-Based Vehicle-Cloud Collaboration Approach for Joint Estimation of State-of-Energy and State-of-Health

**DOI:** 10.3390/s22239474

**Published:** 2022-12-04

**Authors:** Peng Mei, Hamid Reza Karimi, Fei Chen, Shichun Yang, Cong Huang, Song Qiu

**Affiliations:** 1School of Transportation Science and Engineering, Beihang University, Beijing 100191, China; 2Department of Mechanical Engineering, Politecnico di Milano, 20156 Milan, Italy; 3School of Transportation and Civil Engineering, Nantong University, Nantong 226019, China; 4BYD Auto Industry Company Limited, Shenzhen 518118, China

**Keywords:** joint estimation, state of energy, state of health, vehicle-cloud collaboration

## Abstract

The state-of-energy (SOE) and state-of-health (SOH) are two crucial quotas in the battery management systems, whose accurate estimation is facing challenges by electric vehicles’ (EVs) complexity and changeable external environment. Although the machine learning algorithm can significantly improve the accuracy of battery estimation, it cannot be performed on the vehicle control unit as it requires a large amount of data and computing power. This paper proposes a joint SOE and SOH prediction algorithm, which combines long short-term memory (LSTM), Bi-directional LSTM (Bi-LSTM), and convolutional neural networks (CNNs) for EVs based on vehicle-cloud collaboration. Firstly, the indicator of battery performance degradation is extracted for SOH prediction according to the historical data; the Bayesian optimization approach is applied to the SOH prediction combined with Bi-LSTM. Then, the CNN-LSTM is implemented to provide direct and nonlinear mapping models for SOE. These direct mapping models avoid parameter identification and updating, which are applicable in cases with complex operating conditions. Finally, the SOH correction in SOE estimation achieves the joint estimation with different time scales. With the validation of the National Aeronautics and Space Administration battery data set, as well as the established battery platform, the error of the proposed method is kept within 3%. The proposed vehicle-cloud approach performs high-precision joint estimation of battery SOE and SOH. It can not only use the battery historical data of the cloud platform to predict the SOH but also correct the SOE according to the predicted value of the SOH. The feasibility of vehicle-cloud collaboration is promising in future battery management systems.

## 1. Introduction

The two significant energy and environmental pollution issues have increasingly become the focus of attention [1]. Automobiles are an essential part of the industrial field, and researchers are also looking for more environmentally friendly energy sources to replace conventional petroleum [2,3,4]. Due to the advantages of high energy density, the high number of cycles, and pollution-free, Lithium-ion battery is widely applied in many industrial fields, such as new energy vehicles and hybrid energy ships [5]. Lithium-ion batteries play an essential role in these fields, and their safety and reliability should be guaranteed. Therefore, the estimation of battery status should be real-time and accurate. The short driving range is the fatal weakness of EVs, and how to improve accuracy is an urgent research hotspot. Battery state estimation is also a significant factor in driver mileage anxiety, related to many factors, such as weather, road conditions, and temperature. Accurate estimation of SOE effectively alleviates range anxiety as it is straightly related to energy consumption [6]. Moreover, battery performance decreases over time due to irreversible physical and chemical alters until the battery is retired [7]. Thus, it is also necessary to accurately estimate battery SOH to guarantee the performance and safety of the battery.

### 1.1. Review of SOE Estimation Methods

SOE is defined as the ratio of the residual energy storage in the battery; the fully charged battery SOE is 100%. Accurate SOE provides the fundamental basis of energy management, load balancing, and security for complex energy systems [8]. The mathematics of SOE can be presented as follows [9]:(1)$$SOE_{t}=SOE_{t_{0}}+\frac{\int_{t_{0}}^{t}P_{\partial }d_{\partial }}{E_{N}}$$
where $SOE_{t}$ and $SOE_{t_{0}}$ denote the SOE values at the time t and the initial state, respectively. $E_{N}$ is the battery nominal energy, $P_{\partial }$ means the battery power at time *∂*.

To date, wide-ranging attempts have been made to improve SOE estimation effectiveness continually. Battery SOE estimation is mainly divided into the direct calculation approach, adaptive algorithm-based approach, and machine learning method [9]. Usually, the most used direct methods are the power integration and mapping methods. Barai et al. [10] propose the power integration method for SOE estimation, and the result shows it is practical to reduce the computational burden. However, this approach automatically results in accumulated errors subject to uncertain noises and measurement faults. The usual mapping method is applied to SOE estimation to avert this problem [11]. Even though the mapping-based method shows function improvements compared to the power integration method, it requires precise equipment and extensive testing. In this case, many scholars handle nonlinear systems using the adaptive algorithm-based approach to solve accumulated errors and device defects. Furthermore, efficient SOE estimators are built to obtain promising results, such as Kalman filtering [12,13], adaptive unscented Kalman filtering [14], extended Kalman filtering, and particle filtering [15]. Given the battery management system’s limited storage and computing performance, many complex algorithms and models in the battery management system (BMS) are challenging to calculate.

With the hot development of machine learning, especially the advent of Alpha Go and AlphaGo Zero, deep learning and reinforcement learning in various fields have been promoted [16]. Meanwhile, deep learning technology is also being studied in battery SOE estimation. For instance, Back Propagation Neural Network (BPNN) was used to catch the battery’s nonlinear and coupling characteristics considering irreversible energy losses from joule heating and electrochemical reactions. Liu et al. [17] take battery temperature, open-cycle voltage, and current as input training of BPNN to overcome the complex electrochemical principles inside the battery. Then the test and simulation results show that BPNN can improve the accuracy and reliability of SOE estimation. Ma et al. [6] propose an LSTM model with additional convolutional layers to enhance the accuracy of SOE prediction. However, the accuracy of the machine learning approach is exceptionally reliant on the amount and quality of the training data and the fitting algorithms. Due to the insufficiency of existing onboard vehicle control unit hardware, machine learning techniques cannot be implemented on the vehicle [18,19].

### 1.2. Review of SOH Estimation Methods

SOE describes the battery energy change state from the microscopic level on a relatively small scale. In contrast, the battery SOH represents the battery state from the macroscopic level on a large time scale. With the charged and discharged battery in real applications, the irreversible and inevitable electrochemical reactions will cause performance degradation. The SOH is frequently used as an index to quantify the aging degree of a battery [20]. The SOH is usually used to reveal the capacity fade [21] or the power fade [22]. The capacity fade presents the component’s loss of capacity, and the power fade denotes the internal impedance augmentation. The mathematic definition of SOH can be given as follows [9]:(2)$$\mathrm{SOH}=\frac{C_{a}}{C_{r}}\times 100\%$$

The mathematical definition of SOH can also be presented as follows:(3)$$\mathrm{SOH}=\frac{R_{a}-R_{r}}{R_{r}}\times 100\%$$
where $C_{r}$ and $C_{a}$ express the rated and actual value of capacity, respectively. $R_{r}$ and $R_{a}$ denote the rated and actual value of internal resistances, respectively.

Accurate estimation of SOH is beneficial to guarantee the electric system running safely and conduce to a better cognition of the degradation rules. Furthermore, it is helpful for precisely estimating the SOE. Many scholars have done much research to improve the accuracy of SOH. Similar to SOE estimation, there are mainly two categories to estimate SOH: adaptive filtering and data-driven [20]. Yan et al. [23] propose a Lebesgue-sampling-based extended Kalman filter to estimate SOH, and the estimated SOH is considered the initial capacity for SOC estimation. The result demonstrates the excellent performance of this method. Xu et al. [24] introduce an adaptive dual extended Kalman filter algorithm to predict SOH and SOC jointly. Vichard et al. [25] build an electrical circuit model and then use an optimization method and a Kalman filter based on an experimental dataset to estimate SOH.

The above methods are practical, but their accuracy needs to be improved compared to machine learning. On the other hand, Kaur et al. [26] propose three different deep learning algorithms, feed-forward neural network (FNN), LSTM, and CNN, to estimate SOH considering external complexity affection. Subsequently, they proved the better performance of FNN. However, they neglect to combine these methods to achieve better results. In this case, Fan et al. [27] introduces a hybrid neural network, GRU-CNN, concerning voltage, current, and temperature to estimate battery SOH. Then the open-access dataset is adopted to prove the effectiveness of GRU-CNN. Inspired by this approach, it is still necessary to adjust the parameters and structure to improve the performance of the neural network algorithm. Furthermore, deep learning is challenging to apply to vehicles due to BMS computing power and storage data limitations.

### 1.3. Key Contributions

Owing to the rise of vehicle-to-everything (V2X) communication, cloud computing has attracted widespread attention from researchers. Combined with the cloud, machine learning also has been reborn in the automotive field. For this reason, this paper proposes a vehicle-cloud collaboration approach for the joint estimation of SOE and SOH via deep learning. Specifically, the significant contributions are summarized as follows:A vehicle-cloud collaboration model is developed to estimate battery state online.A joint estimation of battery SOE and SOH based on deep learning is proposed.SOH is the feedback of SOE estimates for higher accuracy.Macro and micro dimensions of time are used to analyze SOH and SOE.

### 1.4. Paper Organization

The rest of this paper is organized as follows. Section 2 proposes a collaborative strategy containing SOE and SOH estimation in EVs and clouds. Section 3 introduces the datasets and methodology for battery state estimation. In Section 4, the developed model is validated under various working conditions. Finally, a brief conclusion about this paper and future work will be proposed in Section 5.

## 2. Vehicle-Cloud Collaboration

As stated, the vehicle-cloud collaboration strategy integrates multi-model adaptation and machine learning for SOE and SOH joint estimations. Accurate battery models are the basis of battery state estimation algorithms. As to battery management applications in EVs, we established a high-precision battery model, and reasonable computing power requirements are integral to BMS development. The model’s performance will significantly affect BMS’s safety, accuracy, and robustness. The current models for state estimation and behavior analysis of lithium-ion batteries can be roughly divided into electrochemical models, equivalent circuit models (ECM), and data-driven models [28]. Electrochemical models usually contain multiple partial differential equations, and some model parameters are difficult to obtain. Although many methods, such as the average electrode model, simplify the issue, applying them in real-time on EVs is still challenging [11]. ECM uses a variety of circuit elements, such as resistors, capacitors, and voltage sources, to simulate the electrochemical dynamics of lithium-ion batteries. The model parameters have a specific physical meaning, usually a centralized parameter model with few parameters. Due to its excellent combination with various advanced control algorithms, it is widely used in real-time control and simulation [29].

Data-driven algorithms can extract critical information from lithium-ion battery operating data and historical data, and there is no need for in-depth research on the battery reaction mechanism [30]. So it has received increasing attention in the automotive field. However, high-performance data-driven algorithms mostly rely on cloud platforms and big data technologies, which is difficult to reach the requirement through BMS [31]. With the revival of V2X, the limitation of BMS computing power in EVs will be solved by utilizing cloud computing. This paper proposes a learning-based vehicle-cloud collaboration approach combining the advantages of BMS and cloud computing. Specifically, an ECM is established at the EVs to estimate the battery state; a data-driven model is also built in the cloud platform, given in Figure 1. Besides, this paper aims to jointly predict the SOE and SOH of EVs based on the vehicle-cloud collaboration approach under the following basic assumptions.

The cloud platform can store a large amount of battery history data.When the EVs are driving in the networked road environment, they can obtain the networked information in real-time.The communication problem of vehicle–cloud collaboration approach is not considered.

Dissemination of the results in this paper mainly aims at verifying the rationality of the subsequent experiment; in this case, the complexity of the actual conditions is simplified in a targeted manner. After completing the simplified problem, these assumptions will be gradually removed for a more in-depth discussion.

### 2.1. Power Battery Modeling

#### 2.1.1. ECM

Previous research has established that the second-order RC model performs better than the higher-order RC model considering complexity and accuracy [16]. Motivated by this, we take the second-order ECM as the research object. The second-order ECM model consists of three parts, the voltage source, the ohmic resistance, and the RC network, given as Figure 2. Furthermore, the mathematical model of the second-order ECM can be expressed as follows,
(4)$$\left[\begin{array}{c} {\dot{V}}_{1}\\ {\dot{V}}_{2} \end{array}\right]=\left[\begin{array}{cc} \frac{-1}{R_{1}C_{1}} & 0\\ 0 & \frac{-1}{R_{2}C_{2}} \end{array}\right]\left[\begin{array}{c} V_{1}\\ V_{2} \end{array}\right]+\left[\begin{array}{c} \frac{1}{C_{1}}\\ \frac{1}{C_{2}} \end{array}\right]I$$
(5)$$U=U_{OC}-IR_{0}-V_{1}-V_{2}$$
where $U_{OC}$ denotes the open circle voltage, $R_{0}$ means the battery ohmic resistance; $V_{1}$ and $V_{2}$ represent the polarization voltage across the $R_{1}C_{1}$ network and $R_{2}C_{2}$, respectively.

#### 2.1.2. Parameter Identification

Before the ECM model performs SOE estimation, it is also necessary to identify each parameter in the ECM model. This step is related to the accuracy of SOC estimation. Many universities and research institutions have done many experiments on batteries before, such as NASA [32], the University of Maryland Center for Advanced Life Cycle Engineering (CALCE) Battery Research Group [33], and the University of Oxford’s Battery Intelligence Laboratory [34]. Note that these data have been open-sourced, which is helpful for data-driven research methods. To obtain accurate battery parameters, we designed an experimental scheme for the pulse discharge of the 18650 lithium battery in an incubator. A stable temperature discharge test platform is established, as shown in Figure 3. These data, including the OCV-SOC mapping test and model parameter identification test, can be obtained in our test platform. Based on this, particle swarm optimization (PSO) algorithms are used to optimize parameter identification results. The basic parameters of the battery are given in Table 1.

The velocity update equation of the PSO algorithm is shown in Equation (Equation 6); the content distribution on the right side has the following physical meanings: inertial velocity, moving to the individual historical optimal solution and moving to the group optimal solution. The position update equation of the PSO algorithm is shown in Equation (Equation 7):(6)$$v_{k+1}=\omega_{i}v_{k}+b_{1}m_{1}\left[p_{\mathrm{opt},k}-x_{k}\right]+b_{2}m_{2}\left[g_{\mathrm{opt},k}-x_{k}\right]$$
(7)$$x_{k+1}=x_{k}+v_{k}$$
where *v* is the velocity; *x* denotes the position of the particle; *k* is the current iteration number; $\omega_{i}$ represents the inertia weight; $b_{1}$ and $b_{2}$ mean the learning rate factors; $m_{1}$ and $m_{2}$ are the random numbers between 0 and 1; $p_{opt}$ and $g_{opt}$ denote the optimal solution of the particle itself and the global optimal untie, respectively.

The incremental current and open-circuit voltage (OCV) data at 25 °C are analyzed based on the PSO method. The parameters identification results and OCV-SOE are presented in Figure 4 and Figure 5. It can be seen from Figure 4 that the results of PSO optimization basically coincide with the experimental data, which proves the rationality of the PSO algorithm. The fitting data curve of battery SOE and OCV is given in Figure 5. Moreover, this paper uses the polyfit function that comes with Matlab software to fit the experimental data. Among them, polyfit is a polynomial fitting function based on the least square method; the specific curve fitting equation is shown in Equation (Equation 8):(8)$$y=8.86x^{5}-27.74x^{4}+33.68x^{3}-19.41x^{2}+5.83x+2.94$$
where *y* denotes the battery voltage, and *x* means the battery SOE.

### 2.2. Neural Network for SOE and SOH Estimation

Neural networks have good mapping ability for nonlinear systems and have the advantages of model independence and parameter self-learning. Due to the cloud platform having good computing power, neural networks are used in the cloud to build lithium-ion battery models.

#### 2.2.1. Recurrent Neural Network

Compared with BP and CNN, RNN considers the input of the last moment and gives the network the memory function of the previous content. The network will memorize the previous information, and the nodes between the hidden layers are connected. Furthermore, the hidden layer’s input is not only determined by the output of the input layer but also by the output of the hidden layer from the previous moment. As shown in Figure 6, the RNN hierarchical structure mainly consists of the input layer, hidden layer, and output layer, which can be represented as follows:(9)$$S_{t}=f\left(UX_{t}+WS_{t-1}+b\right)$$
(10)$$O_{t}=g\left(VS_{t}+c\right)$$
where $X_{t}$, $S_{t}$ and $O_{t}$ denote the value of input layer, hidden layer, and output layer, respectively; *f* and *g* are the activation functions, while *b* and *c* mean the biases value; *U*, *W*, and *V* represent the weight matrices.

#### 2.2.2. Long Short-Term Memory

Typically, LSTM is an improved RNN that outperforms traditional RNNs in dealing with long-term sequence problems. It adds a state-memory cell to carry information across multiple time steps. LSTM achieves the protection and control of information through three primary structures: the input gate, forget gate, and output gate, respectively. It can be shown in Figure 7. Specifically, the forget gate can decide what information to discard from the cell state, as shown in Equation (Equation 11). The input gate can determine the amount of new information added to the cell state and can be presented as Equations (Equation 12) and (Equation 13). Furthermore, the output value can be defined by the output gate as Equations (Equation 14) and (Equation 15).
(11)$$f_{t}=\sigma_{g}\left(W_{f}x_{t}+W_{f}h_{t-1}+b_{f}\right)$$
(12)$$i_{t}=\sigma_{g}\left(W_{i}x_{t}+W_{i}h_{t-1}+b_{i}\right)$$
(13)$$c_{t}=f_{t}\cdot c_{t-1}+i_{t}\cdot \sigma_{c}\left(W_{c}x_{t}+W_{c}h_{t-1}+b_{c}\right)$$
(14)$$o_{t}=\sigma \left(W_{xo}x_{t}+W_{ho}h_{t-1}+b_{o}\right)$$
(15)$$h_{t}=o_{t}\cdot \sigma_{g}\left(c_{t}\right)$$
where $f_{t}$, $i_{t}$, and $o_{t}$ denote the forget gate, input gate and output gate, respectively; $h_{t}$ and $c_{t}$ are the hidden state and cell state, respectively.

#### 2.2.3. Bi-Directional Long Short-Term Memory

The Bi-LSTM network was proposed to access the information of the input sequence in both forward and backward. The forward LSTM and backward LSTM are combined with Bi-LSTM; they can offer extra context information and has strong learning ability. The structure of Bi-LSTM is given in Figure 8. This module uses Bi-LSTM to learn the capsule layer’s output, which can improve the network features’ fitting effect and the generalization ability on the new data set.

The softmax function is often used as the last activation function of a neural network to normalize the output of a network to a probability distribution over predicted output classes, based on Luce’s choice axiom. Therefore, as to the output of Bi-LSTM, the softmax activation function is used for classification, as shown in Equation (Equation 16).
(16)$$P=softmax(W_{c}s+b_{c})$$

#### 2.2.4. Convolutional Neural Network

CNN has made significant progress in many fields, such as face recognition and autonomous driving. As the data have inherent dependencies between adjacent dimensions, CNN can utilize the share convolving filters to extract the local features; As an artificial neural network, the structure of CNN can be divided into three layers: the convolutional layers, pooling layer, and fully Connected Layer, given in Figure 9. Among them, the convolution layer includes the convolution kernel, convolution layer parameters, and convolution kernel. The convolutional layer, called data features, extracts the input information, such as battery current, voltage, and temperature. Each data reflect these features in a combined or independent way. The primary role of the pooling layer is to downsample without damaging the recognition results. The fully connected layer is mainly used for classification. The features obtained through the convolution and pooling layers above are classified at the fully connected layer. The weight of each neuron feedback is different according to the weight; then, the classification result is obtained by adjusting the weight and the network.

#### 2.2.5. Bayesian Optimization

One critical aspect of our design is the selection of the configuration parameters of the proposed network (Bayes-Bi-LSTM). Instead of applying the traditional manual-based tuning of the model parameters, this paper adopts a probabilistic Bayesian framework through which the model configuration parameters are optimally tuned.

Bayesian optimization is used to find the best hyperparameters for the Bi-LSTM network. Specifically, Bayesian optimization generates a set of hyperparameters that produce the lowest loss function (in our case, RMSE) by evaluating the previous results. The possibility of finding the best hyperparameters increases with the number of optimization iterations. Hyper-parameter optimization problem can be given as follows:(17)$$X^{*}=\arg\max_{X\in U}f\left(X\right)$$
where $X^{*}$ denotes the set of optimal parameters, *U* means the candidate sets, f(x) is the lowest loss function. The Bayes’ theorem used in the Bayesian optimization process is given as follows [35]:(18)$$P\left(f|D\right)=\frac{P(D|f)P(f)}{P(D)}$$
where *D* means the set of observations, *f* denotes the unknown objective function, P(f) is the prior of the marginalized *f*. Besides, $D=\left\{\left(x_{1},y_{1}\right),\left(x_{2},y_{2}\right)\cdots \left(x_{n},y_{n}\right)\right\}$.

#### 2.2.6. Joint Estimation for SOE and SOH

Kim et al. [36] propose a CNN-LSTM neural network that can effectively extract spatial and temporal features to predict energy consumption. Motivated by the above method, this paper combines the advantage of Bayes-Bi-LSTM and CNN-LSTM to estimate SOH and SOE jointly. Specifically, the framework of the joint estimation method proposed in this paper is shown in Figure 10. First, the vehicle-end power battery communicates with the cloud platform in real-time to obtain the battery’s current, voltage, and temperature data at the present moment. In this case, the value of SOE is roughly estimated by the CNN-LSTM method. Then, the Bayes-Bi-LSTM algorithm calculates the SOH according to the battery charge and discharge times recorded in the cloud. Finally, the estimated value of SOH is used as the input value of CNN-LSTM to correct the SOE.

## 3. Datasets and Methodology for Battery Estimation

After analyzing the neural network algorithm, we need to select an open-source dataset to analyze the aging characteristics of the lithium-ion battery. Therefore, this section mainly introduces the datasets used in deep learning, and then test schemes are proposed. In addition, a dynamic stress test (DST) is processed in our battery test platform to prove the effectiveness of the proposed method.

### 3.1. Description of Datasets

In order to make this study more general, the Battery Data Set collected by the National Aeronautics and Space Administration (NASA) Ames Prognostics Center of Excellence (PCoE) [37,38] is chosen for the constructed model’s training and validation. The NASA battery set contains multiple sets of data, and this paper selects the Battery Aging ARC-FY08Q4 data set, including B0005, B0006, B0007, and B0018 four battery life test cycles. In the experiment, the battery type is 18650 Li-cobalt cells; room temperature is set at 24 °C. During battery charging, the battery is charged in constant current mode at 1.5 A until the battery voltage reaches 4.2 V. Then charge in constant voltage mode until the charge current drops to 20 mA. During the battery discharge process, the battery is discharged in a constant current mode of 2 A until the voltages of the batteries B0005, B0006, B0007, and B0018 drop to 2.7 V, 2.5 V, 2.2 V, and 2.5 V, respectively, and then stop discharging. The condition for stopping the experiment was a 30% drop in the cell SOH of the battery. The detailed battery parameters can be found in Table 2. Moreover, to further verify the reliability of the proposed deep learning algorithm, we conducted a DST test based on the battery test platform. The DST test is applied to simulate the actual operating condition for EVs, controlled by discharge power instead of a constant discharge rate. This makes the testing procedure more complicated and closer to real operational situations.

### 3.2. Methodology

First, we extract the B0005, B0006, B0007, and B0018 four-cycle datasets from the NASA open-source dataset. For battery SOH, the first three cycles are taken as the training set, and the B0018 is considered the test set to predict the accuracy of battery SOH. We first predict the battery SOH based on the battery historical data information on the cloud platform. Due to the previous and future information that Bi-LSTM can make good use of, we establish a Bi-LSTM model for SOH prediction. As the model parameters largely determine the algorithm’s quality, we use the probabilistic Bayesian framework to optimize and adjust the model configuration parameters. In the case of getting the SOH prediction result, we take the current, voltage, and temperature as the input of deep learning, take the SOE as the output and test them in B0005, B0006, and B0007, respectively, to verify the validity of the model. Specifically, the CNN-LSTM is implemented to provide direct and nonlinear mapping models for SOE. These direct mapping models avoid parameter identification and updating, which are applicable in cases with complex operating conditions. It is worth noting that in this process, SOH is also considered as the input of CNN-LSTM. Finally, we consider the SOH as the Influencing factor and compare the SOE accuracy in SOH predicted or without SOH.

The accuracy of predicting value is evaluated by mean absolute error (MAE) and root means square error (RMSE), which are defined as:(19)$$MAE=\frac{1}{n}\sum_{i=1}^{n}\left|f_{i}-y_{i}\right|$$
(20)$$RMSE=\sqrt{\frac{1}{n}\sum_{i=1}^{n}{\left(f_{i}-y_{i}\right)}^{2}}$$
where *n* denotes the number of charging and discharging cycles, $f_{i}$ and $y_{i}$ means the actual value and predicted value, respectively.

## 4. Tests and Results

### 4.1. SOH Estimation Results and Discussion

In order to find the optimal solution for Bi-LSTM parameters, this paper uses the Bayesian optimization algorithm to build an effective Bi-LSTM model. The battery cycling discharge (B0005, B0006, and B0007 of NASA) data are used as the training data set, and the B0007 discharge data are used as the testing data set. This paper sets the Bayesian optimization total iter count ten times. The hyperparameter setting and Bi-LSTM model parameter optimization can be seen in Table 3 and Table 4, respectively. The steps adopted in this study are shown in Figure 11. The optimization of hyperparameters in deep learning can be expressed in Equation (Equation 17), which is to find the combination of hyperparameters that minimizes the model’s generalization error. Bayesian optimization can evaluate the next hyperparameter based on known hyperparameters and model errors. Therefore, Bayesian optimization can fully use known information, search for hyperparameter combinations more efficiently, and make it easier to achieve global optimum. First, we need to process the battery data to avoid the impact of random errors on the experiment. Then analyze the error between the test data and the Bi-LSTM prediction, and iterate through the Bayesian optimization algorithm until the error reaches the expected value. Finally, the structure of Bi-LSTM is adjusted according to the parameters optimized by the Bayes algorithm. The minimum target value between an observed value and the estimated value is given in Figure 12. It can be seen that the minimum target declines with the function count increase, and the observed value, as well as the estimated value, reaches the minimum value in the eighth operation. The relationship between the target value and the predicted value can be seen in Figure 13. The dotted line represents the fitted straight line of the predicted data, and the pink line represents the actual data. We can obtain that the fitted line of the predicted data is basically consistent with the actual value.

It can be concluded that the error is controlled within the desired range; the distribution area of error is mainly in 3%, of which the proportion within 2% reaches 90%, the error distribution histogram as detailed in Figure 14. The SOH estimation results for B0007 in the whole cycle is illustrated in Figure 15; also, the SOH prediction error is given in Figure 16. The proposed SOH prediction method provides accurate SOH estimates compared to the actual SOH curve. Indeed, the most significant estimation error among the result is about 0.06. Near 90% of the error of these points is within 0.02, which further proves that the proposed Bayes-Bi-LSTM prediction method can deal with the dynamic discharging process, which also illustrates the robustness of the proposed method.

### 4.2. SOE Estimation Results and Discussion

In order to verify the accuracy and reliability of the model, we estimate SOE in B0005, B0006, and B0007, respectively. Based on B0005 of the NASA date set, the comparison result of CNN-LSTM and LSTM is given in Figure 17; the tenth cycle sample points B0005 are used for the evaluation. It can be seen that the proposed CNN-LSTM method has a good prediction effect, and the RSME reaches 1.61%, while the RSME of the LSTM approach is 2.46%. The error of CNN-LSTM and LSTM is shown in Figure 18. We can obtain that the proposed CNN-LSTM approach stays within a reasonable error range over time; the LSTM method tends to diverge, which will face challenges in practical application.

Besides, we designed the parameters of the CNN-LSTM, as shown in Table 5. For the other two lithium-ion battery cells, the same experiments were conducted. AS to B0006 of the NASA date set, the comparison result of CNN-LSTM and LSTM is given in Figure 19; the error of CNN-LSTM and LSTM is shown in Figure 20. Furthermore, based on B0007 of the NASA date set, the comparison result of CNN-LSTM and LSTM is given in Figure 21; the error of CNN-LSTM and LSTM is shown in Figure 22. The specific comparison data can be found in Table 6. According to the previous statement, the CNN-LSTM scheme proposed in this paper can improve the shortcomings of LSTM and achieve better prediction results. These results prove that the proposed CNN-LSTM SOE estimation method can achieve high accuracy and robustness for different lithium-ion battery cells. The DST scheme test was carried out to prove further the proposed scheme’s accuracy based on the built battery test platform. The voltage and current change curves under DST conditions are shown in Figure 23, and the duration of the whole cycle is about 4000 s. Likewise, we adopt the proposed CNN-LSTM and LSTM methods for battery SOE prediction. As shown in Figure 24, the battery continues to discharge under DST conditions until the SOE drops to 20%. Among them, the proposed CNN-LSTM method has a better prediction effect, and its RMSE reaches 1.10%, while the LSTM method is only 2.04%.

### 4.3. Comparation

Finally, to compare the impact of SOH prediction on the accuracy of SOE estimation. Consider B0005 as the research object, and take the SOH prediction result as the input of CNN-LSTM. Then comparing the SOH unknown result, the total number of cycles is 180, as shown in Figure 25. It can be seen that in the first 20 cycles, the value of SOE of those estimates with and without SOH correction are almost the same. As the number of battery charges and discharges increases, the estimation accuracy of SOE drops sharply when the SOH is unknown. During the process, the SOE estimation accuracy with SOH prediction results has been maintained well. As the battery SOH is a long process, the change is not evident in the short term. After the battery is used for a long time, the battery SOH will decrease, and the change curve of battery SOE will also change. The original algorithm cannot adapt to the change in SOE. Therefore, it is necessary to predict the SOH to judge the change curve of SOE under a certain SOH.

The drawback in SOE estimation without SOH correction is obvious. Suppose the battery’s capacity cannot be updated, particularly when the battery degrades close to the end of life. In that case, the SOE estimation results may be higher than 10%, which is not usable. Therefore, it is critical to consider the battery SOH or degradation states for SOE estimation. In this case, the accuracy of SOE estimation can be improved dramatically by combining the correction of SOH.

## 5. Conclusions

A vehicle-cloud collaboration strategy that integrates machine learning is proposed for joint battery estimation of SOE and SOH to avoid the degradation of the information island by a single model for state estimation. This paper takes SOH as a time series prediction problem, using a Bayesian approach to optimize the Bi-LSTM model. Then, CNN-LSTM is implemented to provide direct and nonlinear mapping models for SOE. In the training process, the voltage, current, and temperature are considered as input, and SOE is severed as the output to learn the estimation model. During the testing phase, the SOE is estimated based on the model learned during the training, taking the real-time current, voltage, and temperature as input. In addition, to verify the influence of SOH on battery SOE estimation, we take the predicted value of SOH with current, voltage, and temperature as input and compare it with the model without SOH information. The simulation result shows the proposed Bayes-Bi-LSTM and CNN-LSTM models’ effectiveness. The relative error of SOE and SOH is within the expected accuracy range. Furthermore, it can be concluded from the comparison result that the SOH prediction is critical for the SOE estimation. Overall, this study strengthens the idea that vehicle-cloud collaboration is promising in future battery management. The CNN-LSTM parameters proposed in this paper can be further optimized, and the corresponding conclusions will be given in the subsequent work. In addition, the algorithm of device-cloud fusion depends mainly on the communication signal. The follow-up work will also design the control strategy for the communication problem to optimize the proposed vehicle-cloud collaboration strategy.

## Figures and Tables

**Figure 1 sensors-22-09474-f001:**
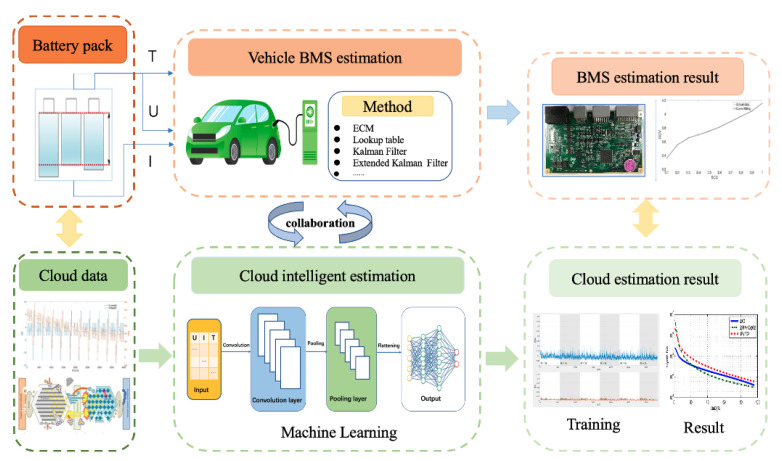
The schematic of vehicle-cloud collaboration.

**Figure 2 sensors-22-09474-f002:**
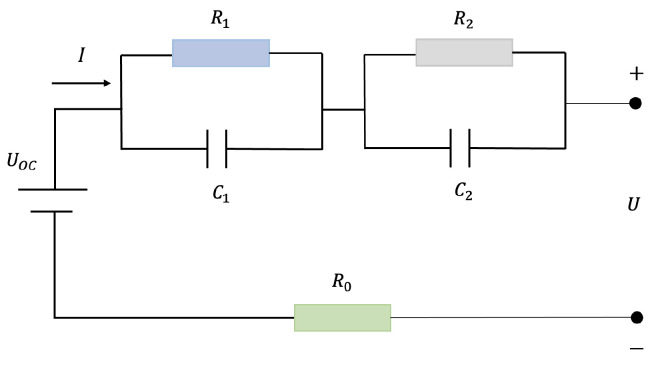
The structure of second-order ECM.

**Figure 3 sensors-22-09474-f003:**
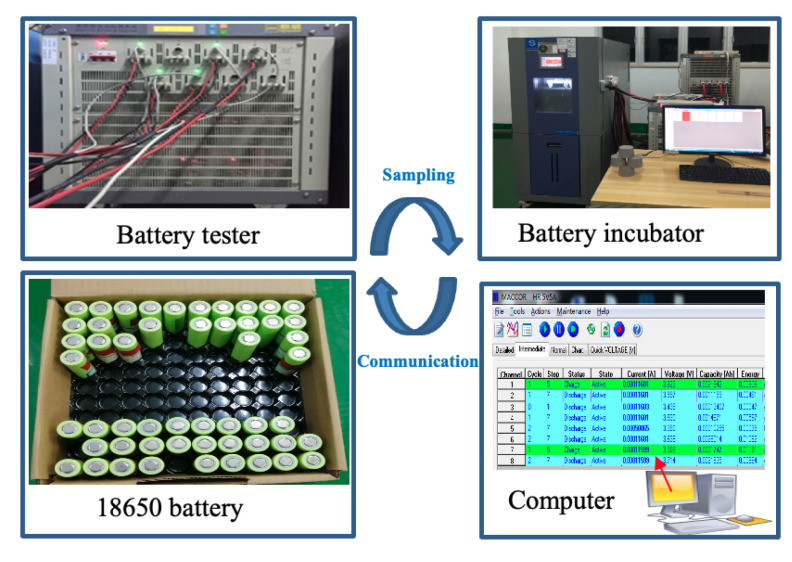
Schematic diagram of the battery experiment platform.

**Figure 4 sensors-22-09474-f004:**
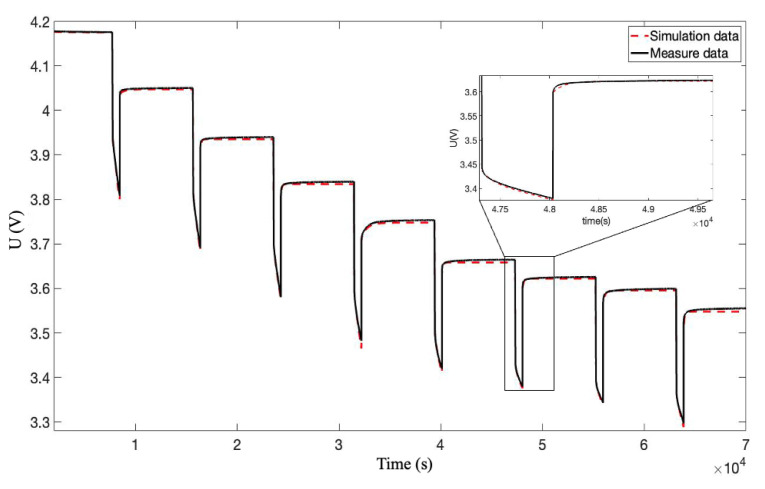
Pulse discharge test and parameter fitting.

**Figure 5 sensors-22-09474-f005:**
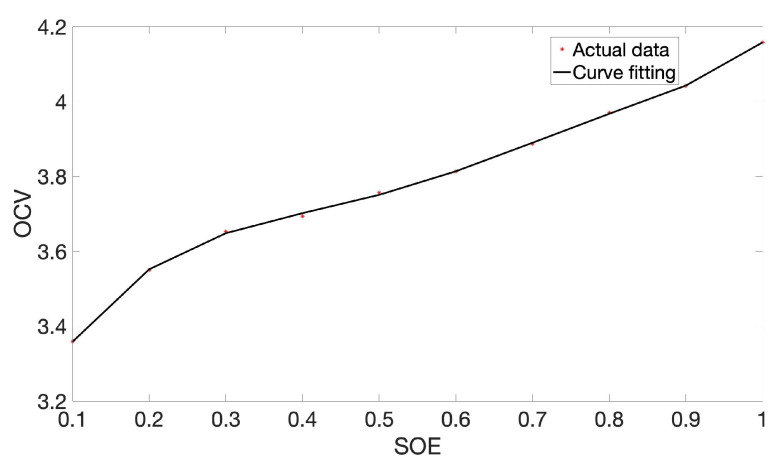
OCV-SOE curve.

**Figure 6 sensors-22-09474-f006:**
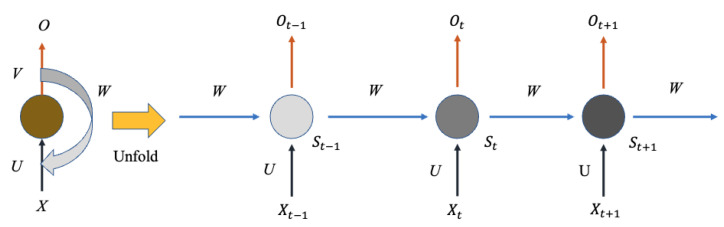
The structure of RNN.

**Figure 7 sensors-22-09474-f007:**
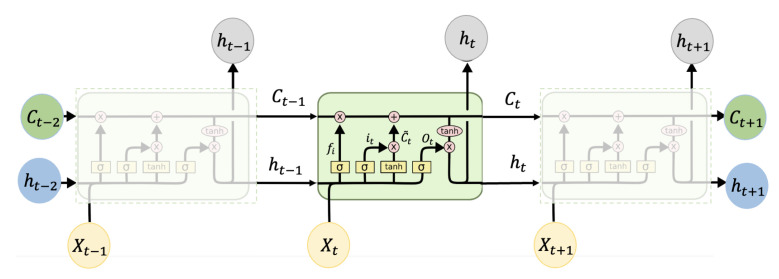
The hierarchical structure of LSTM.

**Figure 8 sensors-22-09474-f008:**
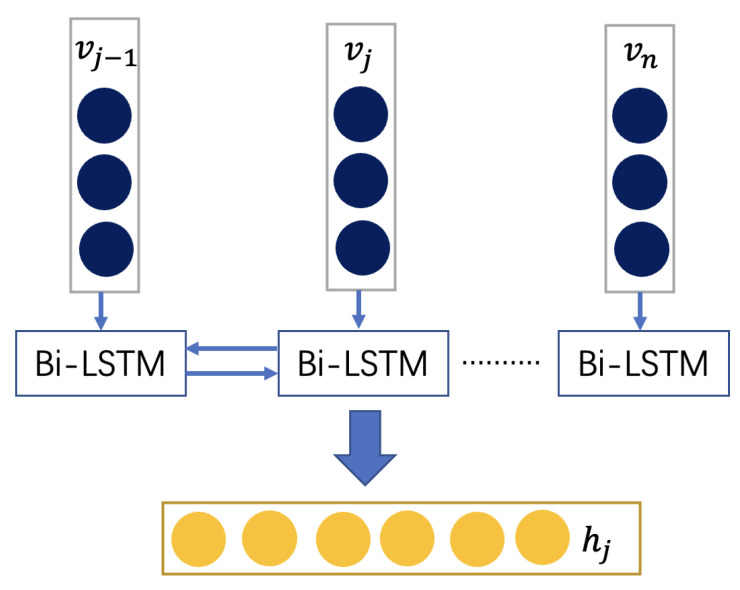
The schematic of Bi-LSTM.

**Figure 9 sensors-22-09474-f009:**
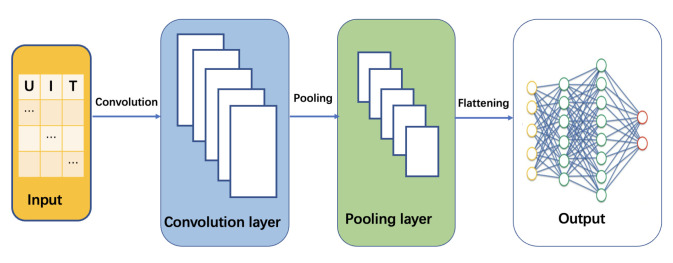
The structure diagram of CNN.

**Figure 10 sensors-22-09474-f010:**
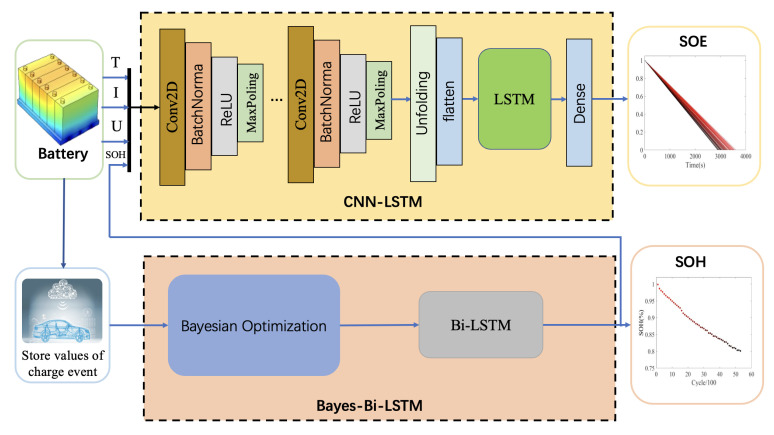
The schematic of joint estimation approach.

**Figure 11 sensors-22-09474-f011:**
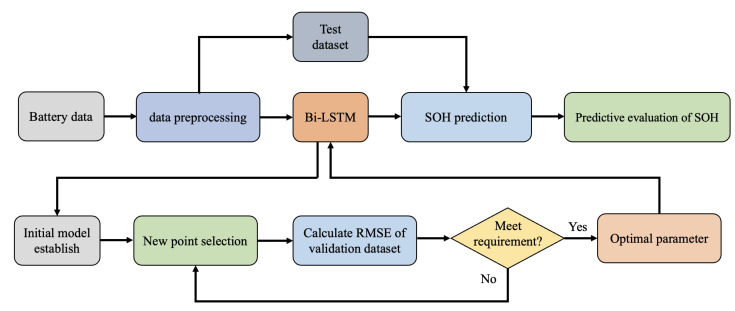
Flowchart of the proposed Bayes-Bi-LSTM.

**Figure 12 sensors-22-09474-f012:**
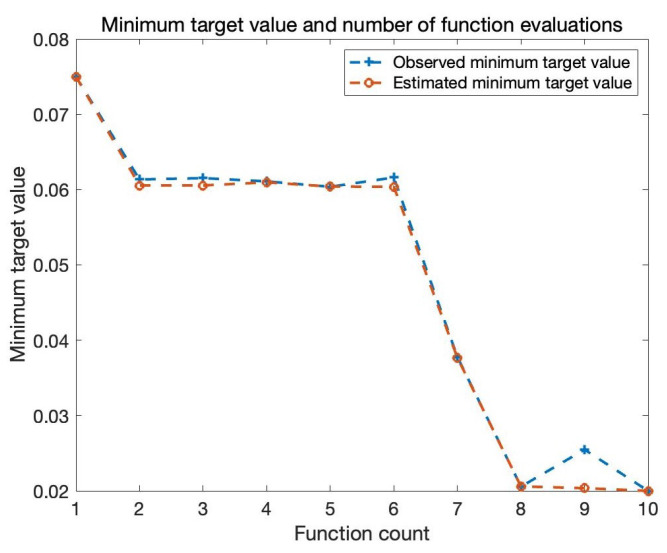
The minumum target value between observed value and estimated value.

**Figure 13 sensors-22-09474-f013:**
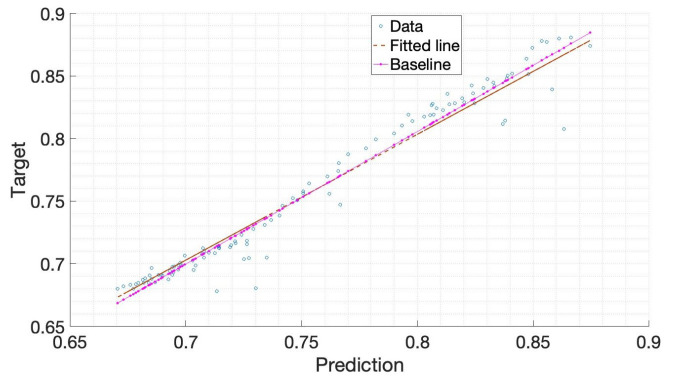
The regression line of output and real value.

**Figure 14 sensors-22-09474-f014:**
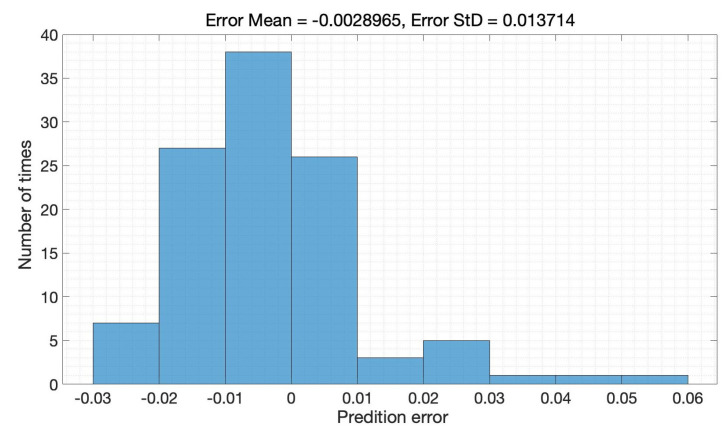
The error distribution histogram.

**Figure 15 sensors-22-09474-f015:**
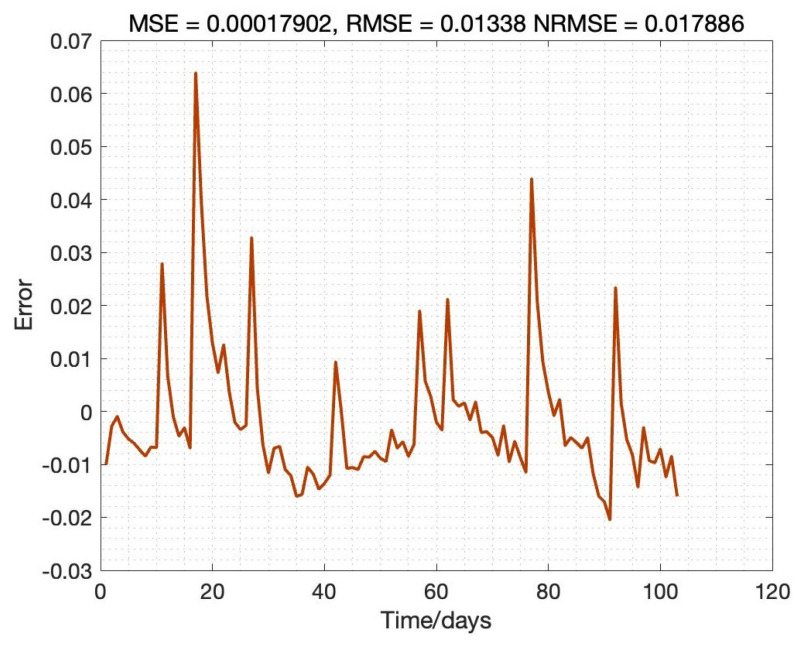
SOH prediction error.

**Figure 16 sensors-22-09474-f016:**
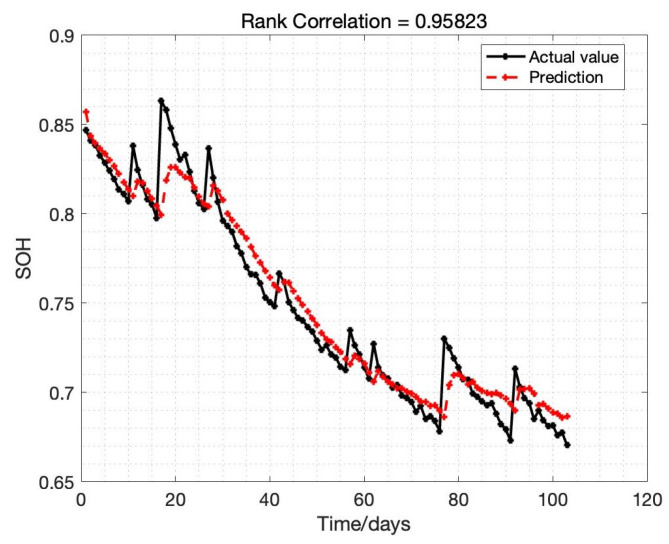
SOH prediction result.

**Figure 17 sensors-22-09474-f017:**
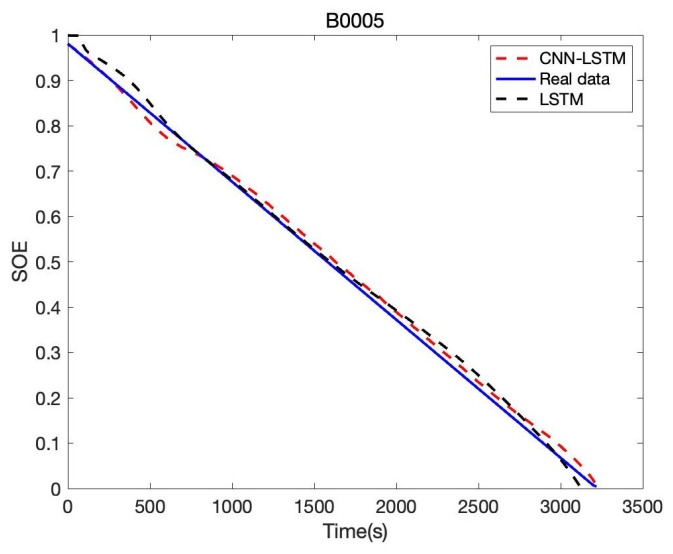
Battery SOE prediction result of B0005.

**Figure 18 sensors-22-09474-f018:**
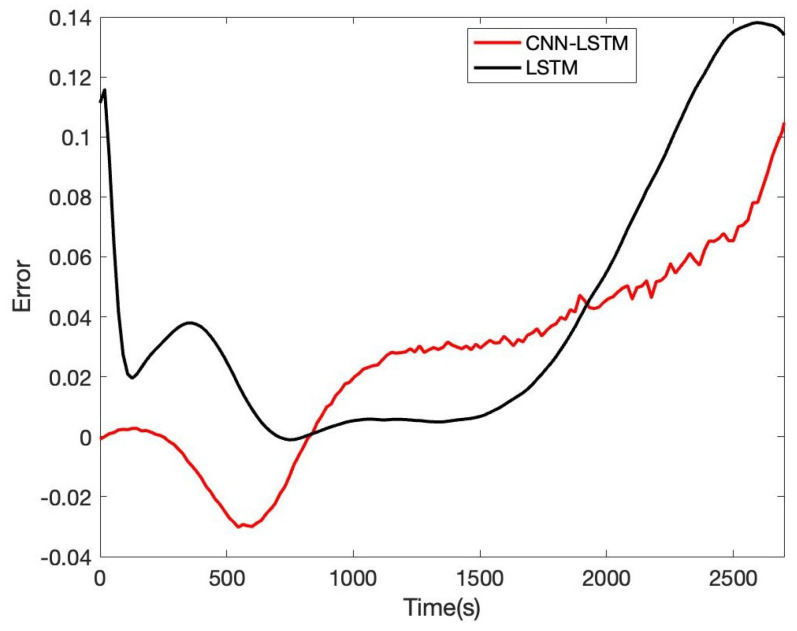
Battery SOE prediction error of B0005.

**Figure 19 sensors-22-09474-f019:**
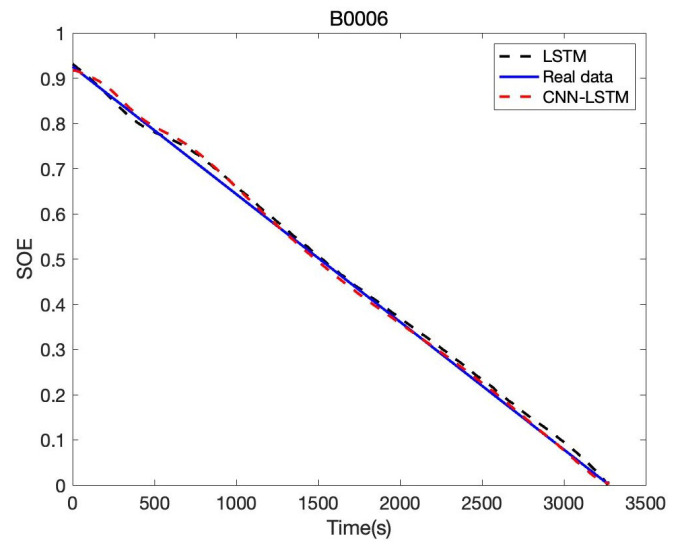
Battery SOE prediction result of B0006.

**Figure 20 sensors-22-09474-f020:**
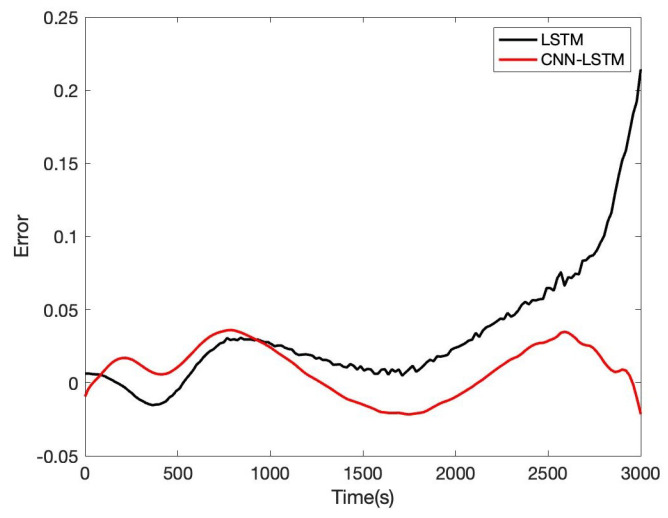
Battery SOE prediction error of B0006.

**Figure 21 sensors-22-09474-f021:**
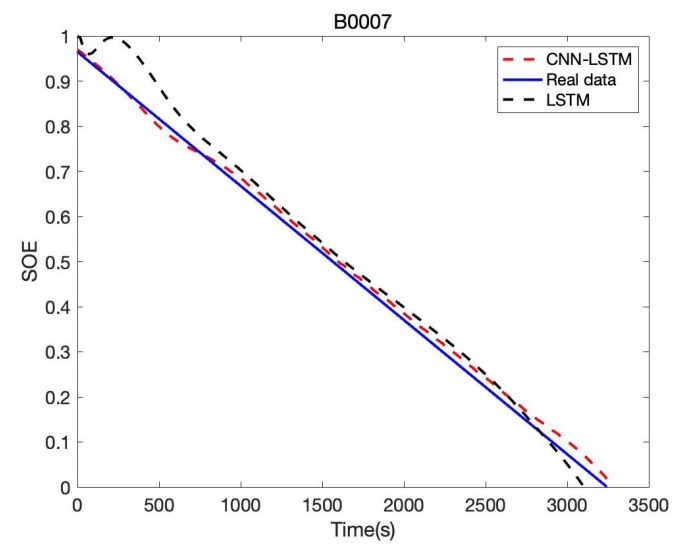
Battery SOE prediction result of B0007.

**Figure 22 sensors-22-09474-f022:**
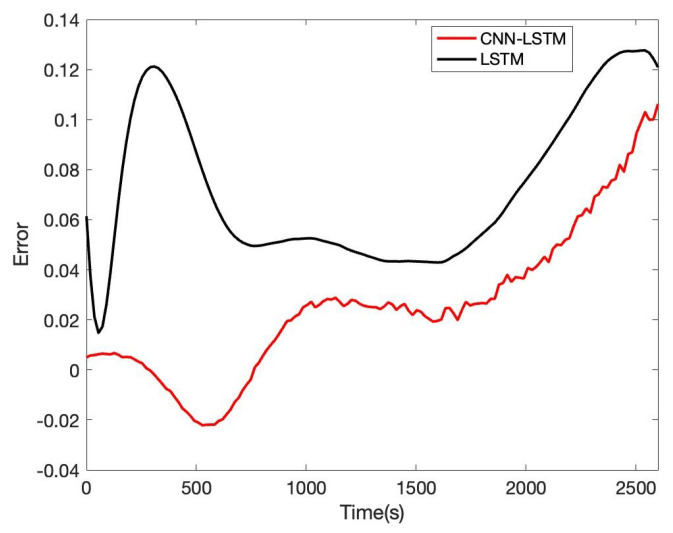
Battery SOE prediction error of B0007.

**Figure 23 sensors-22-09474-f023:**
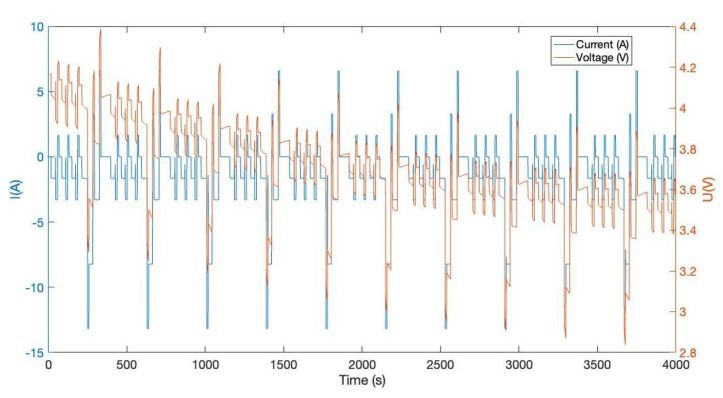
Battery current and voltage changes in the DST condition.

**Figure 24 sensors-22-09474-f024:**
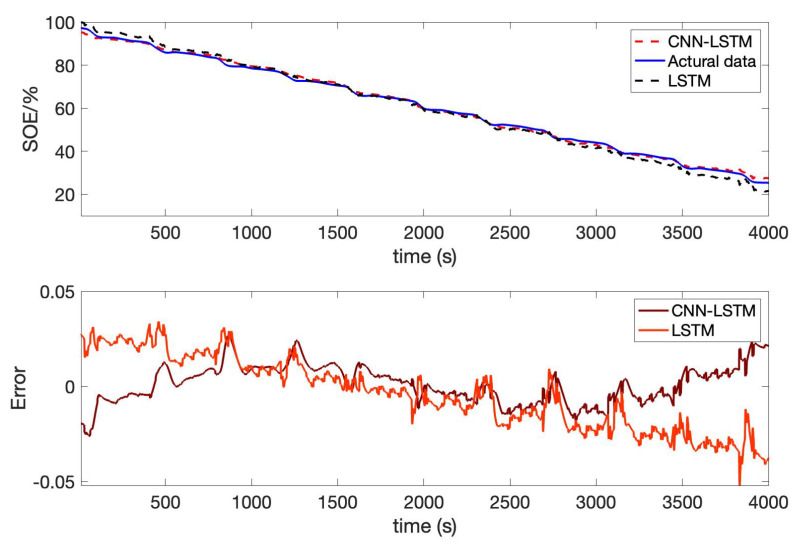
Battery SOE prediction results in the DST condition.

**Figure 25 sensors-22-09474-f025:**
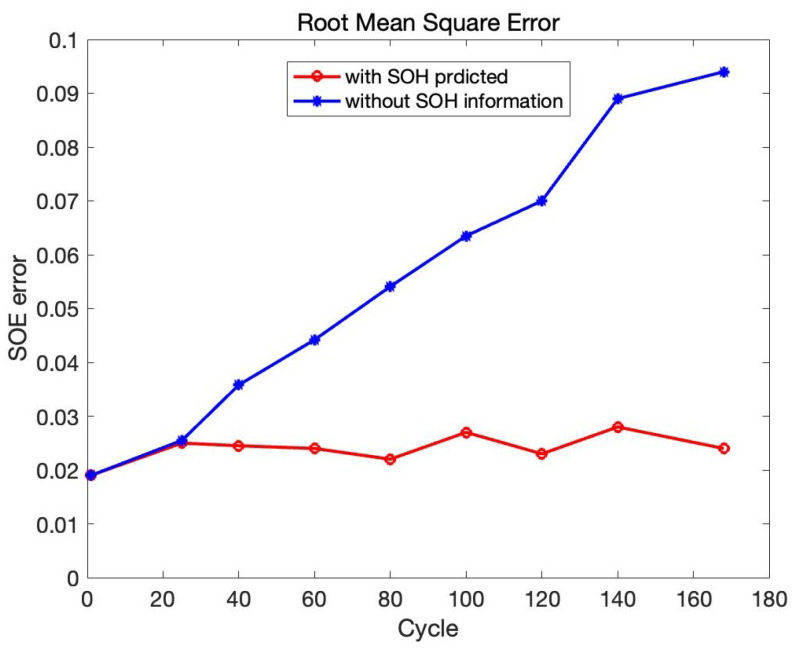
RMSE in B0005.

**Table 1 sensors-22-09474-t001:** Basic specifications of battery.

Type	Nominal Voltage	Nominal Capacity	Upper/Lower Cut-Off Voltage
18650	3.6 V	2.54 Ah	4.2 V/2.5 V

**Table 2 sensors-22-09474-t002:** The battery-specific parameters of the experiment.

Battery Number	Temperature/°C	Rated Capacity/Ahr	Termination Voltage/V	Cycles
#5	24	2	2.7	168
#6	24	2	2.5	168
#7	24	2	2.2	168
#18	24	2	2.5	132

**Table 3 sensors-22-09474-t003:** Hyperparameter setting based on Bayes-Bi-LSTM algorithm.

Hyperparameter	Value
Maximum epochs	10
Minimum batch size	16
Dropout value	0.7
Max itration number	10

**Table 4 sensors-22-09474-t004:** Bayesian optimization.

Iter	Number of Layer	Number of Units	Initial Learn Rate	$L_{2}$ Regularization
1	2	174	0.02042	$2.958\times 10^{-9}$
2	2	200	0.066371	$1.456\times 10^{-6}$
3	3	64	0.054394	$2.3297\times 10^{-8}$
4	1	68	0.44111	$8.3725\times 10^{-5}$
5	3	197	0.9156	$3.3127\times 10^{-3}$
6	1	87	0.095566	$8.8124\times 10^{-7}$
7	1	54	0.0322	$1.0002\times 10^{-10}$
8	1	62	0.01005	$3.8664\times 10^{-3}$
9	4	113	0.010022	$2.7007\times 10^{-5}$
10	1	61	0.25309	$6.6029\times 10^{-7}$

**Table 5 sensors-22-09474-t005:** The proposed CNN-LSTM architecture.

Type	Filter	Kernel Size	Stride	Value
Convolution	32	(10,1)	1	-
Activation (eLu)	-	-	-	-
Pooling	-	(10,1)	2	-
Convolution	32	(10,1)	1	-
Activation (eLu)	-	-	-	-
Pooling	-	(10,1)	2	-
Learning rate	-	-	-	0.001
Minimum Batch Size	-	-	-	30
Maximum Epochs	-	-	-	60
Learning rate drop factor	-	-	-	0.8
Gradient threshold	-	-	-	1

**Table 6 sensors-22-09474-t006:** RSME comparison results.

Method	B0005	B0006	B0007	Battery Test
LSTM	0.0246	0.0118	0.03721	0.0204
CNN-LSTM	0.0161	0.0102	0.0164	0.0110

## Data Availability

No new data were created or analyzed in this study. Data sharing is not applicable to this article.

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
