# Peer review of "A Learning-Based Vehicle-Cloud Collaboration Approach for Joint Estimation of State-of-Energy and State-of-Health"

_sensors, 2022, doi:10.3390/s22239474_

Round 1
Reviewer 1 Report
This paper proposes a joint SOE and SOH prediction algorithm, which combines long short-term memory (LSTM), Bi-directional LSTM (Bi-LSTM), and convolutional neural networks (CNNs) for electric vehicles based on vehicle-cloud collaboration.
In Subsection 1.1, a review of some works with SOE was done and some important consideration was indicated as well as limitations for automotive sector. The same was done for SoH in section 1.2.
Figure 9 is very interesting however its information is not detailed in text.
Figure 12 is not clear. Figure 12 needs to be further explained. In addition it also does not contain details of what each curvature is presented. Absence of captions also occur in Figure 13.
Section 4.2 should have a lot of detail on the analysis of the graphs, however, the detailing of the results is extremely poor in information for the reader.
The author, in general, does not analyze the results obtained and leaves this task to the reader of the article. Sections 3 and 4 are the most relevant and are not detailed.
The author greatly motivated the work involving cloud, insufficient hardware unit to apply machine learning, but did not make this link with the results obtained.
Reviewer 2 Report
This manuscript proposed a vehicle-cloud cooperation strategy combining machine learning for joint battery estimation of SOE and SOH. The strategy can avoid the degradation of the information island prosperity by a single model for state estimation. The experimental results and analysis in the manuscript are also very clear and reasonable. Here are some suggestions for improving the quality of the paper.
1.Linguistics, readability of the paper should be further polished.
2.There are format errors in the paper, such as 59 lines on page 2, Figure 12 on page 10, etc. The authors should check and correct them carefully.
3.There are several contents in the paper that need to be confirmed by the author. Is the description of the SPO method on line 169 on page 5 correct? The meaning of U,W,V and other variables in line 178 on page 6. Should Ct on page 7, line 190, be ct?
4.Please explain the softmax activation function mentioned in line 198 on page 7 appropriately.
5.The discussion of SOE estimation results in Section 4.2 is not detailed enough. It is suggested that the author add relevant explanations to better highlight the research contribution.
6. The dataset selected in this paper is from 2007, and it is recommended to use the new dataset for testing, and the authors should add new experiments to illustrate the frontier of the research.
7.What’s the limitation of your method? Are there other ways that the results can be further improved? In the conclusion, some content about future prospects can be added appropriately.
Reviewer 3 Report
The reviewer has the following comments and suggestions for the authors to be addressed:
- The abstract needs to be revised to include a clear idea and scope of the work. That is a missing part in the abstract. Also, please highlight the contributions and the main findings.
- Line 16: please arrange the keywords alphabetically.
- Lines 59 and 269 have some typos when mentioning reference 15 and “whocle”.
- Subsections 2.2.1 through 2.2.4 require better titles. For example, the whole wording instead of the abbreviations.
- The arrangements and locating of the figures and tables in the manuscript need to be revised. Many of the illustrations are embedded too late when they are discussed. Others are placed in places where they break paragraphs into two segments! For an easier and better reading experience for readers, please fix this issue.
- The Bayesian optimization requires a better introduction and explanation. Whenever dealing with an optimization process/problem, many aspects must be fully described such as the main objective/s, constraints, control variables, solution method, stopping criteria, etc. Unfortunately, not all these aspects are satisfied in this manuscript.
- The results section requires more discussion to cover all aspects of the findings. Moreover, the captions of the embedded figures seem to be not fully descriptive and cover what’s in the figures.
Round 2
Reviewer 1 Report
Section 2.1.2 has been substantially improved. Figura 3 was added including a Schematic diagram of the battery experiment platform.
Section 3 was not good. Little text and not much context. You put the methodology figure at the end. You could put it at the beginning to guide the entire chapter. In addition, it is very poor in details.
In section 4, Figures 11, 12 13 and 14 were explained in the section. The text has also been substantially improved. However, can be improved.
The author greatly motivated the work involving cloud, but still not being convincing in the text link. I suggest improving this cloud context in a final version, as it was placed in the title of the work, so apparently it should have better detail.
Reviewer 2 Report
In this paper, a vehicle-cloud collaboration strategy that integrates machine learning is proposed. After modification by the author, from the overall perspective of the paper, the method of the paper is correct, the content is more substantial, the result is reasonable, but there are still a few small problems need to be modified.
1. There is a format error in the 12th line of the abstract.
2. Please check the grammar and format of the sentences carefully again.
3. The expression of equation (4) is incorrect. The I in the second term on the right of the equation should be extracted separately.
4. It is suggested to select a section of data in Figure 4 to make a local enlarged image.
5. Please explain the method used for curve fitting in Figure 5.
6. In order to better reflect the frontier of research, it is necessary for the author to choose newer data to replace the current data for simulation experiments.
Reviewer 3 Report
The authors addressed all the raised concerns in round 1. No further changes are requested!
Author Response
We are very grateful to the reviewer for your time and careful assessments of our paper.
Round 3
Reviewer 2 Report
In this paper, a vehicle-cloud collaboration strategy that integrates machine learning is proposed. After modification by the author, from the overall perspective of the paper, the method of the paper is correct, the content is more substantial, the result is reasonable.This paper can be accepted in its current form.